# Nanomaterial-Reinforced Portland-Cement-Based Materials: A Review

**DOI:** 10.3390/nano13081383

**Published:** 2023-04-16

**Authors:** Víctor A. Franco-Luján, Fernando Montejo-Alvaro, Samuel Ramírez-Arellanes, Heriberto Cruz-Martínez, Dora I. Medina

**Affiliations:** 1Tecnológico Nacional de México, Instituto Tecnológico del Valle de Etla, Abasolo S/N, Barrio del Agua Buena, Santiago Suchilquitongo, Oaxaca 68230, Oaxaca, Mexico; 2Tecnologico de Monterrey, Institute of Advanced Materials for Sustainable Manufacturing, Monterrey 64849, Nuevo Leon, Mexico

**Keywords:** nanomaterials, rheology, filler effect, pozzolanic reaction, microstructure, mechanical properties, durability

## Abstract

Portland cement (PC) is a material that is indispensable for satisfying recent urban requirements, which demands infrastructure with adequate mechanical and durable properties. In this context, building construction has employed nanomaterials (e.g., oxide metals, carbon, and industrial/agro-industrial waste) as partial replacements for PC to obtain construction materials with better performance than those manufactured using only PC. Therefore, in this study, the properties of fresh and hardened states of nanomaterial-reinforced PC-based materials are reviewed and analyzed in detail. The partial replacement of PC by nanomaterials increases their mechanical properties at early ages and significantly improves their durability against several adverse agents and conditions. Owing to the advantages of nanomaterials as a partial replacement for PC, studies on the mechanical and durability properties for a long-term period are highly necessary.

## 1. Introduction

Cement was discovered 175 years ago. Since then, it has been widely used as a binder material in the construction industry [1]. In the last 65 years, the amount of cement produced has increased 34-fold, and its demand is projected to increase by approximately six billion tons by 2050 [2]. Currently, several types of materials are produced with ordinary Portland cement (PC), such as pastes, mortars, concretes, high-performance concrete (HPC), ultra-HPC (UHPC), self-compacting concretes (SCCs), ultra-high-performing SCCs, and pavement concrete. All these PC-based materials are key components in building construction owing to their advantages, such as low-cost, high heat-storage capacity, high chemical inertia, and ease of molding into various sizes and shapes [3]. To determine the quality and performance of PC-based materials, a wide range of their properties in fresh and hardened states is rigorously evaluated according to several standards such as the American Society for Testing and Materials, Europe Standard, America Concrete Institute, and Brazilian Cement Standard [4] (see Table 1).

Despite the advantages of PC in building construction, its manufacture and use present several problems and challenges. For example, in obtaining PC clinker, excessive raw materials and energy consumption of approximately 2900 to 3300 MJ/tons of clinker are required [5,6,7]. Therefore, the construction industry is responsible for 5–7% of the global emission of CO_2_ into the atmosphere [8]. Moreover, when PC-based materials are used in construction, they are exposed to adverse conditions, e.g., earthquakes, marine environments, and/or industrial zones with high CO_2_ concentrations [9]. These conditions can cause severe damage and problems in PC-based materials, such as delamination, cracking, and spalling of concrete covers, reinforcement corrosion, carbonation and alkalinity loss, decalcification of calcium-silicate hydrate, dissolution of calcium hydroxide, and fire damage [10,11]. The aforementioned conditions can result in high costs of maintenance and restoration as well as deterioration in the mechanical and durability properties, which can induce loss in stability, building collapse, and safety of users [12,13].

The same two-fold interest can be found in the seismic rehabilitation of existing Reinforced Concrete (RC) buildings. In fact, nowadays, the rehabilitation of existing RC buildings by means of optimized calculation methods [14,15] and the use of more performing and thinner concrete jackets [16,17,18,19] can contribute to reducing the production of Portland Cement and to having better-engineered interventions.

To solve the problems and challenges of PC-based materials, several research works have been conducted, including the use of alternative fuels [20] and raw materials to obtain clinker [21], alternative cementitious binder [22,23], and supplementary cementitious materials (SCMs) [24,25,26]. Among the SCMs, nanomaterials can be highlighted.

In recent years, nanomaterials have been widely investigated in different areas such as agriculture, energy storage and conversion, medicine, sensors, electronics, environmental remediation, and construction [26,27,28,29,30,31,32,33,34,35]. The above is due to the fact that the physic-chemical properties of nanomaterials are very different compared with those of bulk materials [36]. In particular, in construction materials, various investigations have focused on the incorporation of distinct types of nanomaterials (e.g., oxide metals, carbon, and industrial/agro-industrial wastes) in the production of PC-based materials to improve their properties in fresh and hardened states compared with PC-based materials without the addition of nanomaterials [37,38,39,40].

In this context, some reviews about the effect of nanomaterials on the mechanical properties or durability of PC-based materials have been published [41,42,43,44,45,46,47]. However, these studies only analyzed a type of nanomaterial or its mechanical or durability properties. Therefore, a detailed review of the properties of nanomaterial-reinforced PC-based materials in fresh and hardened states is required. In the current study, we present an exhaustive analysis of the effect of nanomaterials on the fresh and hardened state properties of PC-based materials.

## 2. Properties of Nanomaterial-Reinforced PC-Based Materials in Fresh-State

### 2.1. Hydration Heat

PC hydration includes the combination of all simultaneous physical-chemical reactions among the PC particles, water, and other additives [48]. In this perspective, Chen et al. [49] reported that the hydration heat of pastes significantly increased when 5% and 10% of PC were substituted by TiO_2_ nanoparticles. Moreover, they demonstrated that hydration heat occurred earlier compared with pastes using PC only. Other studies [50] demonstrated that nanoparticles such as SiO_2_ (0.5–1.5% of PC replacement) and CaCO_3_ (1–3% of PC replacement) also increased the hydration heat. On the other hand, some investigations have mentioned that the addition of Fe_3_O_4_ nanoparticles (0.5–5%) did not affect the hydration heat of the pastes [51,52].

An explanation for the increase in the hydration heat of PC-based materials with the addition of nanomaterials could be due to two factors. First is that nanoparticles act as potential heterogeneous nucleation sites for hydration products, which was corroborated by Senff et al. [37], who identified the rapid growth of crystalline structures in mortars with 2.5% of SiO_2_ nanoparticles at 9 h of age, which was related to the start of the acceleratory stage of PC hydration [53]. The second factor is the chemical reactivity as a dissolution consequence of the nanoparticle pozzolanic activity [50].

### 2.2. Setting Time

When nanoparticles are combined with PC-based materials, the water demand has been reported to increase, and their properties in the fresh state can be modified [45]. For instance, Chen et al. [49] reported that the initial and final setting times of pastes with 5% and 10% of TiO_2_ nanoparticles were shorter than those of the control, and these times decreased with the increase in nanoparticle addition. In another study, Zhang et al. [54] reported that the use of SiO_2_ nanoparticles (2%) in concrete reduced both the initial and final setting times. Further, Ghafari et al. [55] observed the same effect on the setting times when PC was partially replaced with 1–4% of SiO_2_ nanoparticles in the manufacture of UHPC.

### 2.3. Workability/Flowability

Additives have been known to affect the workability of PC-based materials positively or negatively [56]. In the case of nanomaterials, the literature has reported that partial replacement of PC by 5% and 10% of TiO_2_ nanoparticles reduced the workability of mortars that required the use of a superplasticizer (SP) to maintain adequate workability [49]. Comparable results were reported by Joshaghani and Moeini [57], who demonstrated that the consistencies of fresh mortars decreased when 3% and 6% of SiO_2_ nanoparticles were used as a partial PC replacement. In this context, Sobolev et al. [58] observed that 0.21% of SP was needed for each additional 1% of SiO_2_ nanoparticles to manufacture concrete with adequate workability.

Other nanoparticles, such as Graphene Oxide (GO), have also caused workability problems in paste when PC was partially replaced by 0.05%, 0.10%, and 0.20% of this nanomaterial [59]. Similarly, the addition of 0.5%, 0.1%, 1.5%, and 2% of Al_2_O_3_ nanoparticles as a PC replacement decreased the workability of concretes [60]. Nevertheless, Sikora et al. [52] discovered that partial replacement of PC by 1% to 5% of Fe_3_O_4_ nanoparticles did not negatively affect the workability of mortars because these nanoparticles are characterized by low surface area and absence of porous morphology (see Figure 1). Therefore, the demand for water did not increase.

Other studies have investigated the effect of nanoparticles on the flowability of PC-based materials. Li et al. [50] found that the addition of 0.5%, 1.0%, 1.5%, and 2.0% of SiO_2_ nanoparticles to manufacture UHPC decreased its flowability by 20–34%. The same researchers [50] have also reported a similar tendency when PC was substituted by 1%, 2%, 3%, and 4% of CaCO_3_ nanoparticles, where the flowability of UHPC decreased by approximately 23–35%.

Distinct types of nanomaterials, such as TiO_2_, clay, SiO_2_, and ZnO, have also experienced workability/flowability problems in PC-based materials [61,62,63,64]. These workability and flowability problems in PC-based materials were due to several factors, such as high surface area, low particle size, and agglomeration of nanomaterials, which resulted in less free water to contribute to the workability and flowability [65,66].

### 2.4. Bleeding and Segregation

Concretes are composed of several elements with different densities, and in dormant periods, the aggregates settle or segregate, leaving a layer of water over the surface [67]. In this context, limited studies have reported that partial substitution of PC by 0.25% and 2% SiO_2_ nanoparticles increases the cohesiveness of cement paste and thus reduces the bleeding and segregation of concretes [68,69].

## 3. Properties of Nanomaterial-Reinforced PC-Based Materials in Hardened-State

### 3.1. Microstructure

#### 3.1.1. Filler Effect/Pozzolanic Reaction

Several studies have reported that certain types of nanoparticles can or cannot act as pozzolanic material. For example, Chen et al. [49] reported that the relative mass ratios of 5% and 10% of TiO_2_ remained constant on pastes at 3, 7, and 28 days of age. The abovementioned information could indicate that such nanoparticles function as nonreactive fillers. Similarly, Barbhuiya et al. [70] found that the substitution of PC with 2% and 4% of Al_2_O_3_ nanoparticles did not produce a new crystalline phase in pastes at one and three days of age. However, the peaks of Ca(OH)_2_ and gypsum increased at seven days. We need to mention that the formation of Ca(OH)_2_ compounds is a consequence of PC hydration and not from a pozzolanic reaction. Therefore, Al_2_O_3_ nanoparticles do not possess pozzolanic properties, and they act as fillers on the microstructure of pastes (see Figure 2). Similar conclusions have been reported in the literature for other nanomaterials such as CaCO_3_, Fe_3_O_4_, carbon nanotubes (CNTs), and carbon nanofibers (CNFs) [51,52,71,72].

On the other hand, different studies have reported that SiO_2_ nanoparticles possess pozzolanic properties [54,73,74]. Du et al. [75] evaluated the content of Portlandite in pastes with only PC and PC with 0.9% of SiO_2_ nanoparticles at ages of 1, 7, 14, and 28 days. The results indicated that the Portlandite content in pastes without SiO_2_ increased by approximately 22% on the first day and remained constant until 28 days. In pastes with 0.9% SiO_2_ at one day of age, Portlandite reduced its content by approximately 7% compared with the control. Du et al. [75] attributed such reduction in the pozzolanic reaction between the SiO_2_ nanoparticles and Portlandite to its ultrafine particle size and amorphic character. Similarly, Qing et al. [76] concluded that the higher surface area of SiO_2_ nanoparticles is a significant factor in its pozzolanic reactivity and subsequent formation of a substantial amount of C–S–H. In particular, Mandani et al. [73] compared the reactivity of SiO_2_ nanoparticles against silica fume [74] (see Figure 3). They found that pastes with 3% SiO_2_ showed a lower calcium hydroxide than those with silica fume, which resulted in the higher pozzolanic activity of SiO_2_ nanoparticles.

In addition, to SiO_2_, Papatzani et al. [77] found that montmorillonite nanoparticles reacted with Ca(OH)_2_ to form calcium-silicate/aluminate hydrates, which were identified by thermogravimetric analysis, X-Ray Diffraction (XRD), and Chapelle method in pastes at seven days and eight months of ages. Another nanoparticle that has also shown pozzolanic properties is clay because the XRD analysis of pastes and mortars with 0.6% and 3% of this nanoparticle indicated consumption of Portlandita at one day of age, which increased with time [78,79].

#### 3.1.2. Porosity

The microstructure of PC-based materials consists of a complex network of both pores and cementitious compounds. Porosity is a property that significantly influences the mechanical and durability properties of such materials, and nanoparticles are a type of material that can modify the pore network. In this perspective, Chen et al. [49] reported that the addition of TiO_2_ nanoparticles (5% and 10%) reduced the porosity and altered the pore-size distribution of the microstructures of pastes. Joshaghani and Moeini [57] investigated the influence of adding two types of SiO_2_ nanoparticles (one amorphous and one aqueous colloidal solution) on the porosity of mortars at 90 days. The results indicated that mortars manufactured with 6% of amorphous SiO_2_ reduced the total specific pore volume by approximately 29%. In addition, when PC was partially replaced by 6% SiO_2_ in an aqueous colloidal solution, the total specific pore volume was also reduced. However, the performance of SiO_2_ was slightly lower than in the amorphous state, with a percentage reduction of approximately 25%. Other types of nanomaterials, such as GO, have also been demonstrated to reduce the pores in the microstructure of PC-based materials significantly. In this context, the addition of only 0.20% of GO for PC has been reported to reduce the porosity of mortars by approximately 50% compared with those using only PC at 28 days of age [59]. Similar results were reported by Sikora et al. [52], where 3% of Fe_3_O_4_ in mortars at 28 days of age reduced the total porosity to 23.98%, whereas the control mortar had a total porosity of 25.32%.

As previously mentioned, a complex network consisting of both pores with different sizes and diameters forms part of a microstructure. In this context, Urkhanova et al. [80] found that 0.01% of Carbon Nanoparticles (CNPs) reduced the total porosity of concrete by approximately 12%. However, the results of mercury porosimetry indicated that the pore content in the range of 0.1–0.01 μm increased by 11.3%. The aforementioned result could be considered because the size and distribution of pores in PC-based materials significantly influence the penetration and diffusion of aggressive agents, which could affect their durability properties.

In general, the addition of nanomaterials reduces the pores in the cementitious matrix of PC-based materials. Such benefit is due to the filler effect of nanomaterials resulting from their lower size relative to both the PC and aggregate particles. Moreover, the pozzolanic properties of some nanomaterials create C–S–Hs secondary, which also blocks the pores. In this perspective and as previously mentioned, nanomaterials accelerate PC hydration. Therefore, the cementitious compounds precipitate faster in the matrix. The abovementioned condition creates less porous PC-based materials in a short time compared with those manufactured using only PC.

#### 3.1.3. Density

Few studies have analyzed the effect of nanomaterials on the density of PC-based materials. Senff et al. [37] discovered that PC partial substitution by 2.5% of SiO_2_ nanoparticles slightly decreased the fresh apparent density state of mortar from 2.90 to 2.23 g/cm^3^. On the other hand, the addition of 6% of clay nanoparticles increased the bulk density of pastes at all ages of testing, which were 1, 3, 7, 28, 90, and 360 days [81]. Such increases were 5.63%, 5.6%, 5.8%, 5.5%, 5.1%, and 5.3%, respectively, relative to the control paste. Moreover, the bulk densities increased with time by approximately 6.3% from 1 to 360 days. When PC was partially substituted with 0.1% to 0.5% of CNT in the mortars at 28 days of age, their apparent densities were not significantly different relative to the control mortar, which exhibited a density of 2.20 g/cm^3^ [82]. However, another study performed by Hu et al. [83] indicated that the 28-day bulk density of the paste decreased by 2.64% and 5.29% with the addition of 0.05% and 1% CNT, respectively.

The discrepancy between the densities of nanomaterial-reinforced PC-based materials reported in the literature could be due to the difference between the types of analyzed density because one was the fresh apparent density state, and the other was in a hardened state. Another explanation could be the lower density of SiO_2_ (2.62 g/cm^3^) with respect to PC, with a density of approximately 3.1 g/cm^3^ [84]. Therefore, the PC-based materials manufactured using nanoparticles exhibited lower density than those manufactured using PC only. Unfortunately, the authors [83] did not report the density of clay and CNT for comparison with SiO_2_ nanoparticles.

### 3.2. Mechanical Properties

#### 3.2.1. Compressive Strength

The compressive strength (CS) test is widely used as a standard parameter to learn the quality of PC-based materials. Therefore, its analysis must be discussed when new materials with PC are manufactured. The CS of PC-based materials can be influenced by different factors, namely, water/cement ratio, curation type and time, specimen size, PC type, and chemical additives. Table 2 lists normalized CS values to compare the effect of the partial substitution of PC by nanomaterials in different PC-based materials and ages.

In contrast to SCMs with particle size on the order of micrometers, generally, nanomaterials improve the CS of PC-based materials at the early ages, irrespective if these nanomaterials possess pozzolanic properties or not. At 28 days, few nanomaterials produce PC-based materials with lower CS than the control, only 6.4% of CaCO_3_, 3–5% of Al_2_O_3_, 1% of Fe_2_O_3_, and 0.1% of CNT. However, an optimal dosage or age of the test could later solve the lower CS.

In PC-based materials mixed with nanomaterials without pozzolanic properties, the increase in their CS is due to the acceleration of its hydration processes as well as the changes in their microstructure and reduction in their porosity. Thus, the increase in CS at early ages is higher than at long-term ages. Nevertheless, we need to emphasize the beneficial effect of the pozzolanic properties, specifically of SiO_2_ and clay nanoparticles, on CS both in the early and long-term ages.

Despite the best performance in terms of CS of PC-based materials with nanomaterials, many studies [90,91] have reported problems in agglomeration. For instance, Kong et al. [92] mentioned that the CS of concretes was higher when they were mixed with ultrasonic dispersion. Therefore, researchers have introduced dispersible nanomaterials for homogeneous mixing, and the details of this dispersion technique can be found in [93].

#### 3.2.2. Modulus of Elasticity

Regarding the modulus of elasticity (ME), little research has reported the effect of nanomaterials on this mechanical property. In this context, Li et al. [87] showed that the addition of Al_2_O_3_ nanoparticles to replace 5% of PC increased ME of mortars by 135%, 133%, and 139% at 3, 7, and 28 days of age, respectively. When the percentage of substitution increased to 5%, ME also increased by 154%, 241%, and 243% for the previously mentioned ages, respectively. However, when 7% of Al_2_O_3_ replaced PC, the improvement was low.

Another study on nanomaterials relative to ME of PC-based materials was conducted by Konsta-Gdoutos et al. [72], who confirmed that the addition of 0.1% of CNT and CNF in mortars increased their ME by 73% and 27% at three days of age, respectively. Such a beneficial effect was also observed at 7 and 28 days, in which the increase in ME was 67% and 56% for CNT and CNF and 94% and 95% for CNT and CNF, respectively, relative to the previously mentioned ages. Other studies on CNT revealed increases in ME of concretes at 28 days by approximately 15% and 8% when PC was partially substituted with 0.1% and 0.05% CNT, respectively [94].

Partial substitution of PC with 2.5%, 5%, 7.5%, and 10% of SiO_2_ nanoparticles has been shown to linearly increase the ME of concretes at the ages of 7, 14, and 28 days [95]. Beneficial results in ME of concretes have also been reported with the addition of Metakaolin (MK) nanoparticles. However, the optimal dosage found by Aiswarya et al. [96] was approximately 8% because a higher percentage substitution could negatively affect this mechanical property.

#### 3.2.3. Flexural Strength

Different studies have reported the effect of nanomaterials on the flexural strength (FS) of PC-based materials. In the case of the addition of SiO_2_ and CaCO_3_ nanoparticles (2% for both nanomaterials) [50], the results indicated that FS at seven days of UHPC improved by approximately 20% and 28%, respectively. This trend was also observed at 28 days. However, the improvement was slightly lower. Another nanomaterial that demonstrated positive effects on FS of PC-based materials is GO. Long et al. [59] investigated mortars with 0.05–0.20% of GO and found that its FS at 14 and 28 days increased with the addition of such nanomaterial. Moreover, Long et al. [59] reported that the optimum replacement of PC by GO was 0.20%. Comparable results were also reported in pastes where FS at 28 days improved by approximately 21% and 25% with the substitution of PC by 0.4% and 0.8% of GO, respectively [86].

The previously mentioned studies [50,62,86] showed an improvement in the FS of nanomaterial-reinforced PC-based materials. However, despite the advantage of the use of SiO_2_, CaCO_3_, and GO, a negative effect on mechanical properties was observed with the addition of 1–5% of Fe_2_O_3_ nanoparticles. The mortars, at 28 days of age, that used this nanomaterial exhibited a reduction on their FS, and such decrement was high with the increase in Fe_2_O_3_ nanoparticles [52].

#### 3.2.4. Tensile Strength

Morsy et al. [97] used clay nanoparticles as a PC substitute by 2%, 4%, 6%, and 8% for manufacturing mortars, and they found that the tensile strengths (TSs) at 28 days increased by 11%, 22%, 33%, and 50%, respectively. A study on the effect of Fe_2_O_3_ nanoparticles on the TS of mortars conducted by Nazari et al. [98] reported that the addition of 0.5%, 1.0%, 1.5%, and 2.0% of this nanoparticle improved their TS at 7, 28, and 90 days of ages. Moreover, Nazari et al. [98] discovered that the optimal dosage for best performance was 1.5%. In the case of SiO_2_ nanoparticles, Rajkumar et al. [99] evaluated TS of concretes using 1.5% of such nanomaterial and reported an improvement of 149%, 114%, and 45% at ages of 3, 7, and 28 days, respectively.

### 3.3. Durability Properties

#### 3.3.1. Water Absorption Capacity

Aggressive agents in PC-based materials are dissolved in water. Therefore, water absorption capacity is an important property for the durability of such materials. In this perspective, Joshaghani and Moeini [57] reported that the PC partial replacement with 6% SiO_2_ nanoparticles decreased the percentage of water absorption by approximately 19% compared with mortar that used only PC. Similar results were reported by Rajkumar et al. [99], who added 1.5% of SiO_2_ nanoparticles to manufacture concretes that were immersed in water for 30 min, 60 min, one day, two days, and seven days. Concretes with SiO_2_ nanoparticles demonstrated up to 275% lower values of water absorption than the control.

Other types of nanomaterials have been evaluated in terms of water absorption. For example, Oltulo and Şahin [100] reported that the addition of up to 1.25% of Al_2_O_3_ and Fe_2_O_3_ nanoparticles slightly reduced the water absorption of mortars immersed for 180 days. However, a higher percentage of 2.5% in both nanomaterials increased the water absorption of mortars. In the case of concretes with 0.1% of CNP, their water absorption only decreased by 0.2% compared with the concrete that used only PC [80]. Other researchers have reported that a PC substitution higher than 5% of Al_2_O_3_ nanoparticles increased the water absorption of mortars by 0.13% [101]. Slightly beneficial results relative to water absorption were obtained in mortars with clay nanoparticles [79], where the results indicated that partial substitution of PC by 1%, 2%, and 3% of this nanomaterial reduced the water amount by 0.4%, 0.3%, and 0.7%, respectively, compared with the control.

#### 3.3.2. Chemical Attack

Infrastructure constructed using PC-based materials can be exposed to adverse conditions such as sulfates and acid agents, which can chemically attack such materials and cause cracking, strength reduction, softening, permeability reduction, and expansion [102]. In this context, Tong et al. [103] investigated the effect of GO (0.1% by PC) on the CS of mortars immersed in ammonium-nitrate solution at 15%. They found that CSs of mortars with and without GO significantly decreased when these materials were immersed in the solution for five months. However, mortars with GO exhibited a lower reduction in their CS than those that used only PC. In five months, the CS of mortars without GO decreased by 30%, while the CS of those with GO decreased by 21%.

Ghafoori et al. [104] studied the effect of SiO_2_ nanoparticles on the expansion and CS of mortars exposed to sulfate attack in 4, 8, 12, and 26 weeks as well as in 1 and 1.5 years. They discovered that in 4–12 weeks, the expansion of the mortars with 6% of SiO_2_ nanoparticles was slightly higher than that of the control. However, after 26 weeks, the addition of SiO_2_ nanoparticles significantly reduced such expansion. On the other hand, the SiO_2_ nanoparticles did not demonstrate a beneficial effect on the CS of mortars at 26 and 52 weeks of exposure. Other results were also reported by Arel and Thomas [105], who demonstrated that the addition of SiO_2_ nanoparticles reduced the expansion of mortar after 23 weeks of exposure to sulfate.

#### 3.3.3. Chloride Penetration

Chloride ions (Cl^−^) are an aggressive agent that can induce reinforcement corrosion. Therefore, some methods, additions, and materials have been investigated to mitigate the penetration of such agents into PC-based materials. In the case of nanomaterials, He and Shi [106] manufactured mortars with 1% of SiO_2_, Fe_2_O_3_, Al_2_O_3_, TiO_2_, and clay nanoparticles and cured them for 28 days. They reported that all mortars with nanoparticles demonstrated lower apparent Diffusion Coefficients (DC) than the control material. The order of reduction on DC was clay > SiO_2_ > TiO_2_ > Al_2_O_3_ > Fe_2_O_3_ nanoparticles.

Rapid chloride permeability and rapid chloride-migration tests have shown that the addition of equal to or higher than 5% of SiO_2_ nanoparticles to concretes significantly reduced the Cl^−^ penetration into their cementitious matrices [107]. In this case, the tests were conducted at ages 7, 28, and 90 days, and observed beneficial effects were found at early ages. Du et al. [75] found that 0.3% of SiO_2_ in concretes reduced both the Cl^−^ migration and D_C_ by 28.7% and 31% at 28 days, respectively. However, they mentioned that a higher percentage substitution did not show an additional beneficial effect against Cl^−^ penetration because of the agglomeration of SiO_2_ nanoparticles. When the amount of SiO_2_ increased up to 3.8% PC replacement, both Cl^−^ migration and D_C_ of concretes decreased by approximately 56% and 63%, respectively, at 28 days [108]. The effect of MK nanoparticles on the chloride permeability was investigated by Ragab [109], who performed experiments in the laboratory and found that PC-based materials with 5% and 10% of MK significantly improved their resistance to chloride permeability at 7, 28, and 90 days.

#### 3.3.4. Electrical Resistivity

The electrical resistivity (ER) of PC-based materials is an important physical property, and it is defined as the resistance to the flow of electric current through it [110]. Such properties have been studied by researchers to evaluate the durability of PC-based materials because it is strongly related to the porosity, both the network pore and ion connectivity and transport processes of aggressive agents into PC-based materials [110,111].

The effect of nanomaterials on ER of PC-based materials depends on the employed type. For example, Urkhanova et al. [80] reported that ER of concretes with 0.1% and 0.5% of CNP decreased by a factor of two compared with known electrically conductive additives. Similar negative effects have been reported in mortars at 28 days of age, with 0.1% of CNT and CNF [72]. In this case, the ER value of the mortar control was 8.4 kΩ-cm, whereas those mortars with CNT and CNF demonstrated ER values of 6.1 and 7.2 kΩ-cm, respectively. Urkhanova et al. [80] explained that the negative effect of CNP, CNT, and CNF could be due to their low degree of dispersion and agglomeration into cementitious matrices.

With regard to SiO_2_ nanoparticles, Jalal et al. [68] reported that its 2% addition produced a marginal increase in ER of HPSCC at 7 days. However, at 28 and 90 days, the ER of HPSCC increased by approximately 170% and 365%, respectively. Mandani et al. [107] also reported the beneficial results on ER of concretes with 1%, 3%, 5%, and 7.5% of SiO_2_ nanoparticles and with specific surface areas of 100, 200, and 300 m^2^/g. In the current study, the use of SiO_2_ nanoparticles, especially at higher replacement levels and lower surface areas, demonstrated better performance in enhancing ER of concretes than the finer ones.

#### 3.3.5. Reinforcing-Steel Corrosion

Delamination, spalling, and cracking due to reinforcing-steel corrosion affect the strength, durability, and safety of PC-based materials [112]. In this context, Konsta-Gdoutos et al. [72] investigated the effect of CNT and CNF (0.1% by PC) on the corrosion of reinforcing mortars at 28 days of age, which were immersed in 3.5% NaCl solution for 140 days. In their study, the corrosive activity was evaluated using the corrosion potential (*E_corr_*) and corrosion current density (*i_corr_*). The results indicated that the addition of CNT and CNF produced mortars that were more resistant to corrosion by Cl^−^ because, at the end of the immersion period, such mortars exhibited *E_corr_* values of approximately −280 and −350 mV (Ag/AgCl electrode) compared with mortars without CNT/CNF that showed −450 mV (Ag/AgCl electrode) values. The *i_corr_* results agreed with the values of *E_corr_*, where the control mortar showed *i_corr_* values of 1.66 μA/cm^2^, and those with CNT and CNF showed 0.96 and 0.49 μA/cm^2^, respectively. Contrary results were obtained by Camacho et al. [82], who found that the addition of 0.05%, 0.10%, 0.25%, and 0.50% of CNT to reinforcement pastes and immersed in seawater for 250 days demonstrated more active corrosion than the control. For example, pastes with CNT demonstrated *E_corr_* and *i_corr_* values of approximately −500 mV (SCE electrode) and 0.6 μA/cm^2^, respectively. Meanwhile, the control pastes demonstrated corresponding values of −250 mV (SCE electrode) and 0.09 μA/cm^2^.

In the case of SiO_2_ nanoparticles, UHPC with 3% of SiO_2_ nanoparticles was subjected to an accelerated corrosion test until cracking by Ghafari et al. [113] (see Figure 4). They found that the initiation of corrosion in steel rebars and the corrosion rates were delayed with the addition of SiO_2_ nanoparticles. The abovementioned result revealed that the time for UHPC with SiO_2_ nanoparticles to crack was 8800 min compared with UHPC without nanoparticles, whose cracking time was 3600 min.

Nanowaste materials have also been used as a partial replacement for PC. For example, Mostafa et al. [114] substituted PC with 1% of Rice Husk Ash (RHA), Glass Waste (GW), and MK nanoparticles to manufacture UHPC with high-strength steel, which was immersed in 3.5% NaCl solution for 50 days to induce corrosion. The waste nanoparticles showed a decrease in both the anodic and cathodic overpotential as well as in both open-circuit potential (*E_ocp_*) and *E_cor_r* of reinforcing UHPC to lower negative values. Moreover, such concretes exhibited lower *i_corr_* values than the control (31.72, 14.49, 12.68, and 25.36 μA/cm^2^ for the control, RHA–UHPC, GW–UHPC, and MK–UHPC, respectively).

Another mechanism that induces corrosion in PC-based material is carbonation [115,116]. Some studies [117,118] on the effect of 0.5%, 1.0%, and 5% of SiO_2_ nanoparticles have shown that PC-based materials containing such nanomaterial presented more resistance against carbonation in accelerated tests. Nevertheless, in the case of reinforced pastes with 0.05%, 0.10%, 0.25%, and 0.50% of CNT, the carbonation-induced corrosion activity was higher [82]. During the first 100 days of accelerated carbonation tests, pastes with and without CNT showed similar *i_corr_* values of approximately 0.01 μA/cm^2^. However, after 200 days of exposure, the *i_corr_* value of the control paste was 0.1 μA/cm^2^, and the values of pastes with 0.05%, 0.10%, 0.25%, and 0.50% of CNT were between 0.2 and 0.9 μA/cm^2^. This contradictory carbonation results of PC-based materials with SiO_2_ nanoparticles and CNT could be due to the pozzolanic properties of SiO_2_, which reduced the Ca(OH)_2_ content to react with CO_2_ and, therefore, the carbonation [115].

#### 3.3.6. Freeze–Thaw Cycles

León et al. [119] investigated the PC partial substitution by 5% of SiO_2_ and Al_2_O_3_ nanoparticles against freeze-thaw cycles in mortars. The results indicated that in the first 14 cycles, the mortars with SiO_2_ and Al_2_O_3_ nanoparticles showed similar or stronger resistance than the control mortar. After 28 cycles, the mortars with SiO_2_ nanoparticles exhibited the best performance. The combination of SiO_2_ and Al_2_O_3_ nanoparticles (2.5% by nanomaterial) also improved the performance of the mortars. However, the addition of only Al_2_O_3_ nanoparticles negatively affected the resistance against the freeze-thaw cycles of the mortars.

Another study performed by Chu et al. [120] confirmed that the addition of 0.5% of GO in UHPC reduced the mass loss rate to 0.44%, whereas that of the control was 0.79% after 300 of freeze–thaw cycles. With respect to the relative dynamic elastic modulus, it increased from 95.85% to 97.38% with the addition of GO at the same 300 freeze-thaw cycles. Similar beneficial results were obtained by Yu and Wu [121] with a GO dosage of 0.06%.

According to the analysis of several authors [26,122], the better performance of PC-based materials with nanoparticles against freeze–thaw cycles could be due to their nanocore and refinement effects, which reduced the propagation of microcracks in the microstructure.

## 4. Conclusions

The addition of nanoparticles to PC-based materials was reviewed, and the following conclusions were obtained.

Nanomaterials used as PC partial replacement can be classified as organic, inorganic, and industrial/agro-industry wastes. Moreover, such nanomaterials can have or do not have pozzolanic properties.Nanomaterials act as potential heterogeneous nucleation sites to grow cementitious compounds, which increase the heat during the hydration reaction.Nanomaterials have lower particle size and higher surface area than PC. Therefore, its addition to PC-based materials reduces both their initial and final setting times and workability.Nanomaterials provide a filler effect on the cementitious matrix of PC-based materials, which induces the formation and growth of cementitious products. SiO_2_, montmorillonite, and clay nanoparticles have pozzolanic properties and react with Portlandita to create compounds similar to calcium-silicate hydrates.The filler effect and/or pozzolanic properties of nanomaterials reduce the pore network into the cementitious matrix of PC-based materials, which increases the densities of such materials.In general, partial substitution of PC by nanomaterials can significantly improve the CS, ME, FS, and TS of PC-based materials at early and long-term ages. Such improvement is a consequence of the filler effect, nucleation sites, pozzolanic reaction, and microstructure refinement.The ER of PC-based materials is negatively affected when carbon-based nanomaterials are used as partial replacements for PC. However, SiO_2_ nanoparticles have shown a beneficial effect on the ER of PC-based materials after 28 days of age.The addition of nanomaterials significantly increases the resistance of PC-based materials against several adverse agents that deteriorate or negatively affect the durability of such materials.

## 5. Future Directions

Despite the studies that were performed to understand and evaluate the effect of nanomaterials in the fresh and hardened state properties of PC-based materials, several topics need to be further investigated, such as the following:Research on efficient and low-cost methods to prevent the agglomeration of nanomaterials and subsequent workability problems on the fresh-state properties of nanomaterial-reinforced PC-based materials is necessary.SEM and TEM analysis of the cementitious matrix of nanomaterial-reinforced PC-based materials have to be carried out to elucidate the role of nanomaterials on their microstructural properties.The influence of nanomaterials on the long-term mechanical properties of PC-based materials has to be investigatedAn exhaustive evaluation of the durability in long-term ages of nanomaterial-reinforced PC-based materials as well as elucidation of the mechanism involved in such systems, are requiredLife-cycle assessment of nanomaterial-reinforced PC-based materials has to be conducted to determine their effectiveness in the manufacture of cementitious materials on a large scale.Further, increase the performance of concretes to be used for jacketing in the seismic rehabilitation of existing RC buildings by means of PC partial replacement by nanomaterials.Studies about the combination of two or more nanomaterials can be carried out to improve the properties of PC-based materials.

## Figures and Tables

**Figure 1 nanomaterials-13-01383-f001:**
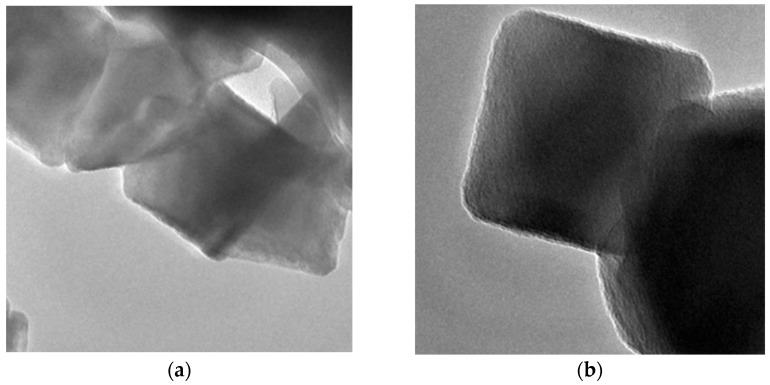
(**a**) Scanning electron microscopy (SEM) and (**b**) transmission electron microscopy (TEM) images of Fe_3_O_4_ nanoparticles. Reprinted/adapted with permission from [52]. 2023, Springer Nature.

**Figure 2 nanomaterials-13-01383-f002:**
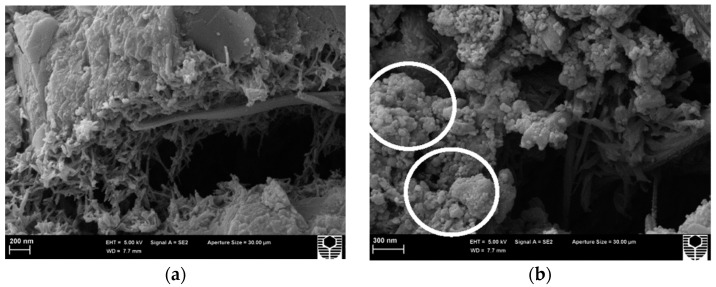
Microstructure SEM images of pastes (**a**) without nanoparticles and (**b**) with Al_2_O_3_ nanoparticles acting as a filler (white circles) Reprinted/adapted with permission from [70]. 2023, Construction and Building Materials.

**Figure 3 nanomaterials-13-01383-f003:**
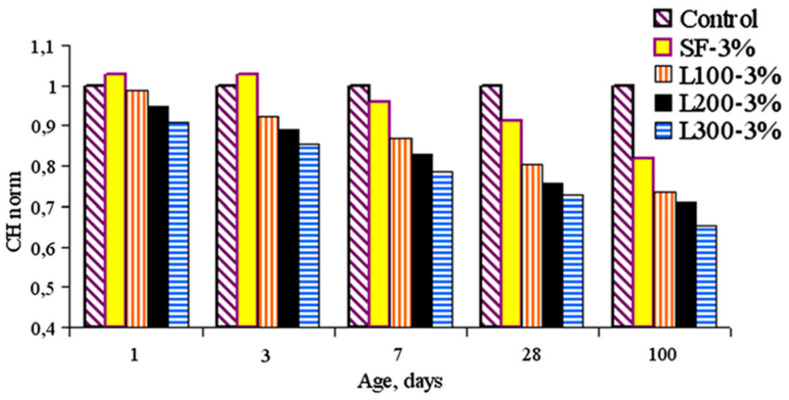
Normalized calcium hydroxide content of paste SF: silica fume and L100, L200, and L300 are nanoparticle SiO_2_ types. Reprinted/adapted with permission from [73]. 2023, Cement and Concrete Research.

**Figure 4 nanomaterials-13-01383-f004:**
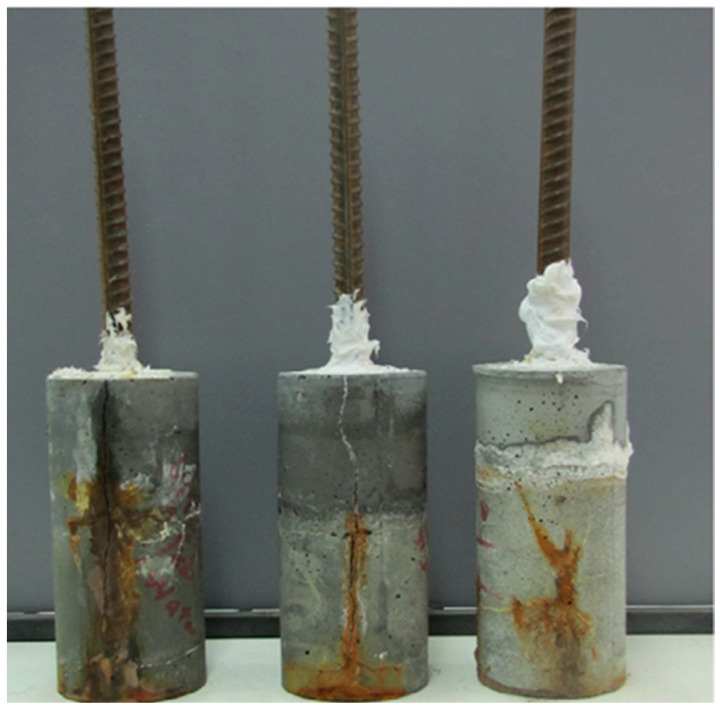
Cracked reinforced UHPC using accelerated-corrosion test Reprinted/adapted with permission from [107]. 2023, Cement and Concrete Composites.

**Table 1 nanomaterials-13-01383-t001:** Properties of PC-based materials.

Properties			
Fresh State	Hardened Sate		
	Microstructural	Mechanical	Durability
Hydration heat Aire content Setting time Workability/Flowability Blending/segregation	Filler effect/ Pozzolanic reaction Porosity Density	Compressive strength Modulus of elasticity Ratio Poisson Flexural strength Splitting strength	Water absorption capacity Chemical attack Chloride penetration Carbonation resistance Electrical resistivity Reinforcing steel corrosion Freeze-thaw cycles

**Table 2 nanomaterials-13-01383-t002:** Normalized CS of PC-based materials manufactured with nanomaterials.

Refs.	PC-Based Material	Nanomaterial	Percentage of Replacement	Normalized CS				
				Days (%)				
				3	7	28	90	360
[49]	Mortar	TiO_2_	5%	127	120	115	—	—
			10%	136	133	123	—	—
[85]	Mortar	TiO_2_	1%	93	97	105	—	—
			3%	89	94	102		
			5%	115	109	128		
[57]	Mortar	SiO_2_	3%	128	115	128	141	—
			6%	138	144	142	152	—
[50]	UHPC	SiO_2_	2%	—	100	104	—	—
		CaCO_3_	2%		121	108		
[71]	UHPC	SiO_2_	0.5%	106	100	105	111	—
			1.0%	109	118	110	112	
			1.5%	113	122	109	105	
			2.0%	104	117	107	98	
		CaCO_3_	1.6%	104	102	109	104	
			3.2%	101	106	110	104	
			4.8%	100	101	110	101	
			6.4%	83	95	94	95	
[59]	Mortar	GO	0.05%	—	—	108	—	—
			0.10%	—	—	112		
			0.20%	—	—	119		
[86]	Paste	GO	0.02%	127	120	114	—	—
			0.04%	127	122	118		
			0.06%	118	118	102		
			0.08%	114	118	109		
[70]	Paste	Al_2_O_3_	2%	104	103	—	—	—
			4%	94	104			
[87]	Mortar	Al_2_O_3_	3%	106	115	93	—	—
			5%	131	105	91		
			7%	93	125	102		
[88]	Paste	Fe_2_O_3_	3%	—	123	126	—	—
			5%		117	115		
			10%		120	104		
	Mortar		3%	—	122	126	—	—
			5%		167	114		
			10%		120	103		
[52]	Mortars	Fe_2_O_3_	1%	—	—	98	—	—
			2%			102		
			3%			121		
			4%			102		
			5%			106		
[80]	Concrete	CNP	0.01%	100	141	138	—	—
			0.10%	105	100	144		
			1.0%	95	94	107		
[89]	Mortar	CNT	0.005%	—	—	104	—	—
			0.02%			111		
			0.05%			102		
			0.1%			96		
[81]	Paste	Clay	6%	125	138	121	104	109
[79]	Mortar	Clay	1%	—	127	113	—	—
			2%		130	113		
			3%		138	125		

## Data Availability

Not applicable.

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
