# Peer review of "Nanomaterial-Reinforced Portland-Cement-Based Materials: A Review"

_nanomaterials, 2023, doi:10.3390/nano13081383_

Round 1

Reviewer 1 Report

Title of the paper: Nanomaterial-reinforced Portland-cement-based materials: A review.

Paper content: The review paper is very interesting and provides a nice overview of the current state of the art concerning the physical and mechanical properties of Portland-Cement-based materials in which a certain percentage of Portland Cement was substituted by a nanomaterial. The list of nanomaterials whose influence is analyzed is thorough and gaps in current research are highlighted. The paper might be suggested for publication, after some minor changes are implemented, as listed hereinafter.

In the introduction, between lines 47 and 59, you explain where the global interest in concretes with partial substitution of Portland Cement by nano-materials originates from, namely:

1)      Necessity to reduce the CO2 emissions, by reducing the production of Portland Cement;

2)      Necessity to reduce the vulnerability of Portland-Cement-based concretes, to actions as Fire, Earthquakes, and aggressive Industrial Environments.

This is very agreeable. However, I would suggest to take into consideration also the good implications regarding the interventions on existing buildings. For instance for the Seismic Rehabilitation of existing RC buildings by means of a) removal of the existing concrete cover, and b) jacketing the RC elements with one of these High Performance and Self Compacting Concretes. The advantage is two-fold:

1)      Money can be saved since the existing RC structures are recovered, thus reducing the need for new buildings and the relevant production of new Portland Cement;

2)      The performance of the rehabilitation intervention by means of RC jackets can be improved, by using one of these new, high performance concretes with partial substitution of Portland Cement with nanomaterials.

In this respect, I strongly suggest you to add two things:

1)      A period in the introduction, appended to the period currently at lines 47-59, with relevant references;

2)      A sentence in the paragraph “Future Directions”.

The former, is as follows:

The same two-fold interest can be found in the seismic rehabilitation of existing Reinforced Concrete (RC) buildings. In fact nowadays, the rehabilitation of existing RC buildings by means of optimized calculation methods (e.g. Bianco and Granati 2015, Bianco et al. 2017) and the use of more performing and thinner concrete jackets (e.g. Tsonos 2008,2010, Shannag et al. 2005, UNI-EN 1504-3:2006) can contribute to reduce the production of Portland Cement and to have better engineered interventions.

Tsonos, A.G., (2008). “Effectiveness of CFRP-jackets and RC-jackets in post-earthquake and pre-earthquake retrofitting of beam-column subassemblages”, Engineering Structures, Elsevier, 30(2008), 777-793.

Tsonos, A.D.G., (2010). “Performance enhancement of R/C building columns and beam-column joints through shotcrete jacketing”, Engineering Structures, Elsevier, 32(2010), 726-740.

Shannag, M.J., Alhassan, M.A., (2005). “Seismic Upgrade of Interior Beam-Column Subassemblages with High-Performance Fiber Reinforced Concrete Jackets”, ACI Structural Journal, Vol.102(1), January-February 2005.

Bianco, V., Granati, S., (2015). “Expeditious seismic assessment of existing moment resisting frame reinforced concrete buildings: Proposal of a calculation method”, Engineering Structures, Elsevier, 101(2015), 715-732. doi:10.1016/j.engstruct.2015.06.047.

Bianco, V., Monti, G., Vari, A., Palmieri, G., (2017). “Seismic amelioration of existing reinforced concrete buildings: strategy to optimize the amount of reinforcement for joints”, COMPDYN2017 – Proceedings of the 6th International Conference on Computational Methods in Structural Dynamics and Earthquake Engineering, 15–17 June 2017, Rhodes Island, Greece, pp. 2823-2842.

UNI-EN 1504-3:2006, “Products and systems for the protection and repair of concrete structures – Definitions, requirements, quality control and evaluation of conformity – Part 3: Structural and non-structural repair”.

The latter is as follows:

Further increase the performance of concretes to be used for jacketing in the seismic rehabilitation of existing RC buildings.

Further Minor Remarks/Comments:

Line 55: I think you should correct “coves” with “covers”. Please check it.

Lines 168-182: Please, consider substituting the word “Portlandita”, which seems Spanish to me, with the corresponding English term “Portlandite”.

Line 190: as in the previous remark.

Line 197: “that can modifies” should be substituted by “that can modify”.

Line 120: “graphene oxide(GO)” should be substituted by “Graphene Oxide (GO)”, which means that, each time you introduce, for the first time, a new acronym, you have to show the words it comes from, using capital letters for the first letter of these words.

Line 215: “of carbon nanoparticles (CNPs)” should be substituted by “of Carbon NanoParticles (CNPs)”.

Line 237: “the bulk densities also increased with of time by approximately..” should be corrected with “..also increased with time by..”.

Header of the fourth column in Table 2: “percentage of replaced” should be substituted by “percentage of replacement”.

Line 304: “..with the addition of metakaolin (MK)” should be substituted by “..with the addition of MetaKaolin (MK)”.

Lines 312 and 319: “..the results indicated that FE at seven..” should be substituted by “..the results indicated that FS at seven..”. Please check it.

Line 371: “..while that those with GO..” should be substituted by “..while CS of those with GO..”. please check it.

Line 388: “..apparent diffusion coefficients (Dc) than..”. to be coherent with what you have done so far, you should name it DC and introduce the Acronym as “Diffusion Coefficient (DC)” and recall it just as DC, afterwards. However, if you meant to label it Dc instead, and want to keep it, you should have introduced it as "Diffusion coefficient (Dc)", which means that just the first letter of "Diffusion" should has been capital.

Line 437: “..-350 mV vs Ag/AgCl..”. Please check if “vs” is necessary.

Line 456: usual problem with first introduction of new acronyms.

Line 530: “..to elucidate the role of nanomaterials their microstructural properties” maybe you meant “the role of nanomaterials and their microstructural properties”. Please check it.

Line 607: the list of authors, in the list of references, cannot end with “et al.”, but you must list the complete name of all authors, regardless of the number of them. Please modify it accordingly.

Line 810: Current 116th reference. Please add the name of the Journal, besides the corresponding DOI, cause it is missing.

Reviewer 2 Report

Responce

on the manuscript Manuscript ID: nanomaterials-2287175 Nanomaterial-reinforced Portland-cement-based materials: A review

Authors: Víctor A. Franco-Luján, Fernando Montejo-Alvaro, Samuel
Ramírez-Arellanes, Heriberto Cruz-Martínez *, Dora I. Medina *

The article present a review of the the properties of fresh and hardened states of nanomaterial-reinforced PC-based materials.  The partial replacement of PC by nanomaterials increases their mechanical properties at early ages and significantly improves their durability against several adverse agents and conditions. Owing to the advantages of nanomaterials as partial replacement for PC, studies on the mechanical and durability properties for long-term period are highly necessary.

The topic of this manuscript is interesting, dealing with analysing and systematization the effect of different nanomaterials on the rheological and hardened state PC-based materials properties. The effect of these nanomaterials  on the properties of PC-based  building materials is still very significant  issue. However, it is a pity that the authors did not systematize the obtained results according to the nature and properties of the odserved nano materials  and this significantly reduces the value of the work.  However some clarifications are needed.

Table1. Capitalize the letter „p“ in the word pozzolanic. In the Table „Carbonation penetration“  as durability properties is pointed , but  this feature is not analyzed further in the text. All the mentioned properties Water absorption capacity , Chemical attack, Chloride penetration, Electrical resistivity, Reinforcing steel corrosion, Freeze-thaw cycles were reviewed in the work. Was there not enough data for this analysis?

 Please correct the word „Portlandita“, that appears throughout the whole text, because in English  it is written as „Portlandite“. In the margin of Fig. 3, please explain  mean  the markings L100, L200, L300, that appear in the figure.

Page 6. The sentence   „ As previously mentioned, a complex network consisting of both pores with different  sizes and diameters forms part of a microstructure. In this context, Urkhanova et al. [74]  found that 0.01% of carbon nanoparticles (CNPs) reduced the total porosity of concrete by  approximately 12%,“    needs a dot, not a comma.

Table2. unspecified unit of measurement for compressive strength.

Page.13. Please think is your spelled "glass wine"   correctly J

As for the conclusions, I miss more detailed analyzes of the used nano materials. Maybe they can be divided into groups - organic, inorganic, with amorphous phase or not, according pozzolanic activity? The combined use of nanomaterials is not mentioned at all.  Because it remains unclear  it is difficult to choose for which purposes specific nanomaterials could be used. 

Round 2

Reviewer 2 Report

The article has been corrected, there are no additional comments

Author Response

We thank to the reviewer for the carefully reading of the manuscript and for all comments and suggestions.